# Psychometric Validation of the Arabic Version of the WPAI:Migraine Questionnaire in Patients with Migraine

**DOI:** 10.3390/neurolint17120202

**Published:** 2025-12-12

**Authors:** Abdulrazaq Albilali, Rema A. Almutawa, Elaf A. Almusahel, Renad A. Almutawa, Nasser A. Almutawa, Faisal M. Almutawa, Shiekha AlAujan, Haya M. AlMalag

**Affiliations:** 1Neurology Unit, Department of Medicine, College of Medicine, King Saud University, Riyadh 11472, Saudi Arabia; 2Department of Neurology, National Neuroscience Institute, King Fahad Medical City, Riyadh 11525, Saudi Arabiaelafalmusahel@gmail.com (E.A.A.); 3Department of Internal Medicine, King Fahad Medical City, Riyadh 11525, Saudi Arabia; renadalmutawae@gmail.com; 4Department of Emergency Medicine, Ad Diriyah Hospital, Riyadh 13717, Saudi Arabia; nasser.almutawa38@gmail.com; 5Department of Medicine, King Faisal Specialist Hospital and Research Centre, Riyadh 12713, Saudi Arabia; almutawafaisal29@gmail.com; 6Department of Clinical Pharmacy, College of Pharmacy, King Saud University, Riyadh 11451, Saudi Arabia; salaujan@ksu.edu.sa (S.A.); halmalaq@ksu.edu.sa (H.M.A.)

**Keywords:** migraine, work productivity, WPAI:Migraine, patient-reported outcome, psychometric validation, Arabic, Saudi Arabia

## Abstract

**Background**: Migraine is a highly prevalent neurological disorder and a leading cause of disability, particularly among working-age adults. Although the Work Productivity and Activity Impairment (WPAI) questionnaire is widely used to assess the functional impact of health conditions, no validated Arabic version specific to migraine is currently available. This study was conducted to validate the Arabic version of the WPAI:Migraine questionnaire among Arabic-speaking migraine patients in Saudi Arabia. **Methods**: A cross-sectional psychometric validation study was conducted at a tertiary headache clinic between June 2023 and January 2024. Adult patients diagnosed with episodic or chronic migraine, based on the International Classification of Headache Disorders, 3rd edition (ICHD-3), completed the Arabic version of the WPAI:Migraine and the validated Arabic version of the Migraine Disability Assessment Scale (MIDAS). Test–retest reliability was assessed after two weeks. Psychometric properties, including reliability, criterion validity, and known-group validity, were evaluated using intraclass correlation coefficients (ICCs), Pearson’s and Spearman’s correlations, and one-way ANOVA. **Results**: Eighty-two patients completed the study (76.8% female; mean age 38 ± 11 years). The Arabic WPAI:Migraine questionnaire demonstrated substantial-to-almost-perfect test–retest reliability (ICC range: 0.68–0.84). WPAI:Migraine domain scores correlated significantly with MIDAS scores—particularly for activity impairment (r = 0.576), presenteeism (r = 0.526), and absenteeism (r = 0.522)—and increased consistently across MIDAS disability grades, supporting validity. **Conclusions**: The Arabic WPAI:Migraine questionnaire is a valid and reliable instrument for assessing work productivity and activity impairment among Arabic-speaking migraine patients, suitable for clinical and research use.

## 1. Introduction

Migraine is one of the most prevalent neurological disorders and the leading cause of disabilities in adults under the age of 50 [1,2]. It can cause a significant burden to different domains of a patient’s life, including the patient’s social life and productivity at work [3]. Migraine peaks during people’s most productive years, showing the highest effect among the age group of 40 to 44 years; as a result, they negatively affect patients’ careers and professional lives [2,4,5]. Thus, the 2019 Global Burden of Disease (GBD) report found migraine to be the leading cause of years of life lost due to disability in working-age women [2,6]. This has also been observed in several geographic regions and countries, with recent reports showing a similarly high prevalence and disability from migraine in the Middle East region, specifically Saudi Arabia [7,8].

In Saudi Arabia, epidemiological studies found the incidence of migraine to be comparatively higher than the global prevalence, with a recent study showing an adjusted one-year prevalence of 25% for migraine [8,9]. Migraine was also ranked as the most common cause of disability in Saudi Arabia between 1990 and 2017 [10]. Several local studies have demonstrated a clear burden of migraine on daily functioning, including reduced work productivity and interruption of routine activities [7,11]. A recent report from the General Authority of Statistics in Saudi Arabia showed that the mean working age group during the first quarter of 2024 was between 25 and 54 years, which also represents the mean age group that suffers most from migraine [8,12]. The most important determinant of migraine burden in the workplace is presenteeism (reduced productivity while at work) rather than just absenteeism (missed work time) [13]. In a large population-based study from the American Migraine Prevalence and Prevention (AMPP) study, Munakata et al. (2009) reported that presenteeism accounted for nearly three times the indirect costs of absenteeism in patients with transformed migraine, highlighting its dominant contribution to productivity loss [14]. Another global survey study across 31 countries found that, in subjects with ≥4 migraine days each month, 52% showed overall work impairment due to migraine, attributed more to presenteeism than absenteeism [15]. Nevertheless, there is a paucity of data in Saudi Arabia and our selected region regarding the impact of migraine and other primary headache disorders on productivity and loss of capability in the workplace, which necessitates the development of a validated and standardized scale to measure the impact of migraine on work productivity.

Multiple patient-reported outcome measures (PROMs) have been developed to assess various aspects of headache-related disability, including the impact of migraine on work productivity [16]. Among these, the Work Productivity and Activity Impairment (WPAI) questionnaire is widely used to evaluate both work-related and non-work-related functional impairment [17]. The WPAI measures the effects of health conditions on productivity, generating scores for absenteeism, presenteeism, overall work productivity loss, and activity impairment outside work [18]. It can be adapted to specific diseases, and the WPAI:Migraine version has been used in several studies to assess migraine-related functional impact [18,19]. Furthermore, the International Headache Society (IHS) guidelines for controlled trials of preventive treatment in episodic and chronic migraine recommend measuring mean changes from baseline using the WPAI [20,21]. Consequently, the WPAI:Migraine has been used in most randomized clinical trials (RCTs) of calcitonin gene-related peptide monoclonal antibodies (CGRP mAbs) to assess headache-related functional outcomes [22,23,24,25]. Although the WPAI has been translated and validated in multiple languages, no data exist regarding the validity of the WPAI:Migraine among Arabic-speaking populations [17,18,26]. Therefore, this study was conducted to validate the Arabic version of the WPAI:Migraine to assess the impact of migraine on work productivity and activity impairment, thereby supporting both clinical management and health economic evaluations among Saudi migraine patients.

## 2. Materials and Methods

### 2.1. Study Design, Participants, and Ethical Considerations

This study employed a cross-sectional psychometric validation design and was conducted between June 2023 and January 2024 at King Khalid University Hospital, King Saud University, Riyadh, Saudi Arabia. The data were collected from migraine patients visiting the headache clinic for new or follow-up appointments. For the study, we recruited adult patients (18 years or older) meeting the International Classification of Headache Disorders, third edition (ICHD-3) [27], criteria for episodic and chronic migraine based on a clinical assessment of one of the investigators, who is a headache specialist (AA), and those who attended their headache clinic appointments. Patients unable to complete the questionnaire or to provide informed consent were excluded. All patients provided written informed consent before answering the questionnaires. The interviewers consisted of medical students, neurology residents, or consultants. The study protocol was approved by King Saud University and the Medical City Institutional Review Board (approved number E-22-7323).

### 2.2. Study Measures and Content Adaptation

The original WPAI was developed to assess work productivity and activity impairment related to general health (WPAI:GH) [28]. The tool has since been translated into many languages, including Arabic for Saudi Arabia [28]. In our study, the WPAI:GH Arabic-Saudi Arabia v2.0 was obtained from the developer Reilly Associates [28]. The questionnaire was slightly modified to adapt it for migraine patients by changing the term “general health” to “migraine”, without altering its structure, scoring, or psychometric foundation. We communicated directly with the developer to ensure the content adaptation process for the Arabic WPAI:Migraine (see Appendix A) was carried out correctly.

The WPAI:Migraine is a six-item questionnaire with a 7-day recall period. The tool includes questions on employment status, hours missed from work due to migraine, hours missed for other reasons, hours worked, and the extent to which migraine impacts both work productivity and other daily activities [29]. Based on the responses to the six items, four scores for absenteeism (work time missed), presenteeism (impairment at work), overall work productivity, and non-work-related activity impairment were obtained as indicated in the Statistical Analysis Section.

The Migraine Disability Assessment Scale (MIDAS) is a validated tool designed to quantify headache-related disability [30]. It has been widely used to assess the functional impact of migraine on daily life. The questionnaire consists of five items that measure the number of days in the past three months during which migraine affected work, household responsibilities, and social activities. Based on the total score, disability is categorized into four grades: Grade I (0–5, little or no disability), Grade II (6–10, mild), Grade III (11–20, moderate), and Grade IV (21+, severe). A validated Arabic version of the MIDAS is available and has been used in prior studies in Arabic-speaking populations [31]. Accordingly, its inclusion in this study was appropriate for assessing migraine-related disability and complementing the WPAI:Migraine questionnaire. Permission to use both the MIDAS and WPAI questionnaires was obtained from the original developers before the study began.

### 2.3. Data Collection

Patients who attended the headache clinic and met the inclusion criteria were asked to fill out an informed consent form with their contact information. All recruited patients who agreed to participate in the study were contacted to arrange a Zoom meeting at their convenience. A Zoom interview was conducted by one of the study investigators to collect sociodemographic information, including age, sex, work activities, education, and marital status. More details about headache characteristics were collected from migraine patients, including headache frequency in the last week, headache duration, and headache intensity using the Visual Analog Scale (VAS).

Following the initial assessment, migraine patients were asked to complete the Arabic versions of the WPAI:Migraine and MIDAS questionnaires, which were either shared on screen for the patient to complete or, in cases of technical limitations, read aloud verbatim by the study team without offering interpretation or guidance. Patients selected their answers independently to preserve the self-reported nature of the instruments. They were also informed about the need for a follow-up session after two weeks to complete the WPAI:Migraine again for test–retest reliability analysis.

### 2.4. Statistical Analysis

The sample size was determined based on common recommendations for validation studies, which typically suggest a minimum of 5–10 participants per item for factor structure analysis and reliability estimates. Given the 6-item structure of the WPAI:Migraine and practical considerations during recruitment, a sample of more than 60 participants was deemed appropriate.

All data were coded and entered digitally into a spreadsheet using the IBM Statistical Package for Social Sciences (SPSS) version 28, Armonk, NY, USA. Continuous data were displayed as mean and standard deviation (SD), and categorical data were displayed as a number and percentage. Scoring of the WPAI domains was performed using the standard methodology:
Percent work time missed due to migraine (Absenteeism): Q2/(Q2 + Q4) × 100.Percent impairment while working due to migraine (Presenteeism): (Q5/10) × 100.Percent overall work impairment due to migraine: (Q2/(Q2 + Q4)) + ((1 − Q2/(Q2 + Q4)) × Q5/10) × 100.Percent activity impairment due to migraine: (Q6/10) × 100.

Here, Q2 is hours missed due to migraine, Q4 is hours worked, Q5 is the self-rated effect on work productivity, and Q6 is the self-rated effect on other activities.

Regarding psychometric properties, the terminology used was aligned with the COSMIN taxonomy [32]. The test–retest reliability of the Arabic WPAI:Migraine was assessed using the intraclass correlation coefficient (ICC) between baseline and 2-week scores for each WPAI domain. The ICC values were classified as 0.01 to 0.2 (slightly fair), 0.21 to 0.40 (fair), 0.41 to 0.60 (moderate), 0.61 to 0.80 (substantial), and 0.81 to 1.00 (almost perfect agreement) [33]. A radar chart was used to visually compare the mean WPAI:Migraine domain scores at test and retest, illustrating consistency across domains and supporting the assessment of test–retest reliability.

To evaluate criterion validity—specifically, concurrent validity—the WPAI scores were correlated with the MIDAS using Pearson and Spearman correlations. Known-group validity was assessed by comparing the mean scores for each WPAI domain across migraine patient subgroups defined by MIDAS disability grade (Grade I–IV). A one-way analysis of variance (ANOVA) was used to determine whether differences in WPAI domain scores between the MIDAS severity groups were statistically significant. A *p*-value of <0.05 was considered indicative of statistical significance.

## 3. Results

### 3.1. Baseline Demographic and Disease Characteristics

A total of 86 migraine patients consented to participate in the study, of whom 82 completed the questionnaires, and 4 did not. The majority of patients were female (n = 63, 76.8%) with a mean (SD) age of 38 (11). The mean (SD) scores of the WPAI:Migraine domains, absenteeism, presenteeism, overall work productivity loss, and non-work-related activity impairment were 15.54 (22.83), 48.09 (29.17), 39.42 (25.20), and 58.29 (28.06), respectively. The majority of the patients (64%) had severe disability (Grade IV) on the MIDAS questionnaire. Additional demographic and clinical characteristics are summarized in Table 1.

### 3.2. Piloting, Cognitive Interview, and Proofreading

Piloting and cognitive interviews were conducted in person with 20 migraine patients. The feedback from those patients recommended no changes and no added comments, as they were able to answer the Arabic items of the WPAI:Migraine without any difficulties. Furthermore, no changes were needed to the Arabic version of the WPAI:Migraine following proofreading by the research team.

### 3.3. WPAI:Migraine Psychometric Analysis

#### 3.3.1. Reliability

Of the 82 eligible migraine patients, 43 completed the second WPAI administration. Test–retest reliability was assessed using ICCs, which ranged from 0.68 for overall work productivity loss to 0.84 for presenteeism, indicating substantial-to-almost-perfect correlations across all domains (Table 2). A radar chart comparing the mean test and retest WPAI:Migraine domain scores showed similar trends across all domains, supporting the instrument’s test–retest reliability (Figure 1).

#### 3.3.2. Concurrent Validity and Known-Group Validity

Concurrent validity was assessed by correlating WPAI domains with MIDAS values using both Pearson’s and Spearman’s correlation. The results showed that three items of the WPAI domains were significantly correlated and positively correlated with varying strengths with the MIDAS total score, with only one item (percent overall work impairment due to migraine) not being statistically significant (Table 3). Known-group validity was assessed by comparing the mean scores of each WPAI domain across migraine patient subgroups, as defined by MIDAS disability grades (Grade I to Grade IV). Migraine patients with a MIDAS grade of IV exhibited significantly greater impairment across all WPAI domains, except MIDAS Grade II in the overall work impairment domain, mostly supporting known-group validity (Table 4).

## 4. Discussion

In this study, we assessed the psychometric properties of the Arabic version of the WPAI:Migraine and confirmed its reliability and validity among Saudi patients with migraine. The results support its use as a reliable and culturally appropriate tool for evaluating migraine-related work productivity and activity impairment.

To date, only two studies have evaluated the WPAI questionnaire among patients with migraine [17,18]. The first study, conducted by Ford et al. [17], evaluated the WPAI in patients with episodic and chronic migraine using a post hoc dataset from the CONQUER clinical trial. They reported moderate test–retest reliability, with ICC values ranging from 0.36 to 0.45 across WPAI domains [17]. Additionally, they demonstrated construct validity through significant correlations between WPAI scores and the role function-restrictive and role function-preventive domains of the Migraine-Specific Quality of Life Questionnaire version 2.1 (MSQ v2.1) [17]. The second study, completed by Domingues et al. [18], was conducted in Brazil using a smartphone-based app [18]. The study focused on the validity of the Brazilian Portuguese WPAI:Migraine by correlating it with headache frequency and headache-related disability, as measured using the Headache Impact Test (HIT-6) [18]. They found significant positive correlations between all WPAI domains—including absenteeism, presenteeism, overall work impairment, and activity impairment—and HIT-6 scores, supporting the questionnaire’s validity [18]. In our study, the Arabic version of the WPAI:Migraine showed substantial-to-almost-perfect test–retest reliability, with ICCs ranging from 0.68 to 0.84 across domains, and demonstrated strong concurrent validity through significant correlations with the validated Arabic MIDAS questionnaire. These findings reinforce the psychometric soundness and cross-cultural applicability of the WPAI in assessing migraine-related functional impairment.

In our study, higher MIDAS scores, indicating greater migraine-related disability, were associated with greater work impairment, particularly in absenteeism and presenteeism. Interestingly, while absenteeism and presenteeism were significantly correlated with MIDAS scores, the composite “overall work impairment” score did not show a significant correlation. This may reflect the way the composite score is calculated or variability in how absenteeism and presenteeism contribute among individual participants. Patients with severe disability (Grade IV) reported the highest levels of productivity loss and time missed due to migraine, consistent with previous findings by Wong et al. [34], who demonstrated a similar pattern in employed individuals in Malaysia. However, WPAI scores must be interpreted within a clinical context. Some patients may continue working despite debilitating symptoms—resulting in lower absenteeism but higher presenteeism—while others may prioritize work over non-work activities such as family or social engagements. Therefore, a lower WPAI score does not necessarily reflect lesser disability. Tracking WPAI scores longitudinally, particularly in response to treatment, may offer a more accurate reflection of functional burden. The validated Arabic WPAI:Migraine thus represents a practical and culturally relevant tool for evaluating migraine-related impairment in Arabic-speaking populations.

This study has several notable strengths. It is the first to validate the Arabic version of the WPAI:Migraine in patients with migraine, addressing a critical gap in available PROMs for Arabic-speaking populations. The study followed a rigorous psychometric validation process aligned with internationally recognized standards, including the COSMIN taxonomy. Substantial-to-almost-perfect test–retest reliability was demonstrated, with ICCs ranging from 0.68 to 0.84. Criterion validity was confirmed through statistically significant correlations with the validated Arabic MIDAS, supporting both concurrent and known-group validity. Moreover, the inclusion of real-world patients from a tertiary headache clinic enhances the clinical relevance of the findings, while the assessment of both work-related and non-work-related domains provides a comprehensive evaluation of migraine-related functional impairment.

Despite these strengths, several limitations should be acknowledged. First, although the sample size (n = 82) is sufficient for initial validation, it may limit the generalizability of the findings. Migraine patients were recruited from a specialized headache clinic, which may introduce selection bias toward individuals with more severe or chronic migraine and may not fully represent the broader migraine population. While this supports the questionnaire’s sensitivity in detecting work impairment among severely affected individuals, future research should assess its performance in populations with milder migraine severity, such as in primary care or community-based settings. In addition, most participants were employed and highly educated, and all were recruited from a single tertiary-care center in an urban setting, which may limit perspectives from primary care or rural populations. Second, approximately half of the migraine patients (43 of 82) completed the 2-week retest. Although attrition at this level is not uncommon in psychometric validation studies, it may introduce bias if those who did not complete the retest differed from those who did. Nevertheless, the ICC values indicated substantial-to-almost-perfect reliability, suggesting that the reduced retest sample size did not meaningfully affect the results. Third, due to the study’s cross-sectional design, the WPAI’s responsiveness to clinical change over time could not be evaluated. Future longitudinal studies are recommended to assess the instrument’s sensitivity to clinical improvement following treatment or intervention. Fourth, regarding the remote administration of the questionnaires via Zoom, although care was taken to preserve the self-reporting nature of the instruments, the presence of a researcher during administration may have introduced interviewer or administration-mode bias, potentially influencing participant responses [35]. Fifth, given that 64% of participants were classified as MIDAS Grade IV (severe disability), the psychometric performance of the WPAI:Migraine in this study largely reflects its utility in high-burden clinical populations. Finally, the present study used the existing Arabic version of the WPAI:General Health (Saudi Arabia v2.0) obtained from the original developer, which was adapted to the migraine context by substituting the term “general health” with “migraine.” Although this approach ensured conceptual consistency and preserved the instrument’s psychometric structure, it did not involve a full forward–backward translation process. Nevertheless, piloting, cognitive debriefing, and proofreading with migraine patients indicated that the adapted Arabic WPAI:Migraine was clear and well understood, with no need for further modification. Minor contextual or linguistic nuances specific to migraine may still influence interpretation in other Arabic-speaking populations; therefore, broader testing across diverse regions is recommended to further support external validity and cross-cultural applicability.

## 5. Conclusions

Overall, the findings of this study confirm that the Arabic version of the WPAI:Migraine is a valid and reliable instrument for assessing the functional impact of migraine among Arabic-speaking patients. It demonstrated substantial-to-almost-perfect test–retest reliability and strong criterion validity, as evidenced by significant correlations with the validated Arabic MIDAS. By capturing both work-related and non-work-related impairment, the Arabic WPAI:Migraine provides a comprehensive and culturally relevant measure of migraine burden. Its availability supports clinical assessment, research, and public health initiatives within Arabic-speaking populations and facilitates its integration into future observational studies and clinical trials evaluating migraine and related disorders.

## Figures and Tables

**Figure 1 neurolint-17-00202-f001:**
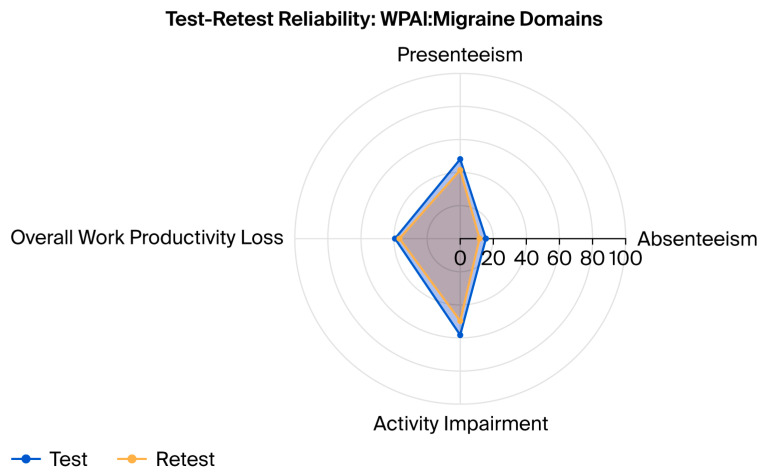
Radar chart comparing mean WPAI:Migraine domain scores at test and retest. The chart illustrates the consistency of responses across the four WPAI:Migraine domains—absenteeism, presenteeism, overall work productivity loss, and activity impairment—demonstrating substantial test–retest reliability among migraine patients (n = 43).

**Table 1 neurolint-17-00202-t001:** Demographic and clinical characteristics of migraine patients.

Characteristics	Migraine Patients (n = 82)
Age, years, mean (SD)	38 (11)
Female sex, (n) %	63 (76.8)
Employment status, n (%)	
Employed	47 (57.3)
Unemployed	35 (42.7)
Education, n (%)	
Low education (illiterate–middle school)	6 (7.3)
High education (high school or above)	76 (92.7)
Headache characteristics, mean (SD)	
Frequency, days/week	3 (2)
Headache duration, hours	18 (30)
Headache intensity, VAS	7 (2)
WPAI:Migraine domains, mean (SD)	
Absenteeism	15.54 (22.83)
Presenteeism	48.09 (29.17)
Overall work productivity loss	39.42 (25.20)
Non-work-related activity impairment	58.29 (28.06)
MIDAS total score, mean (SD)	45.77 (44.50)
MIDAS disability grade (%)	
Grade I	8%
Grade II	9%
Grade III	19%
Grade IV	64%

SD = standard deviation; VAS = Visual Analog Scale; WPAI = Work Productivity and Activity Impairment; MIDAS = Migraine Disability Assessment. WPAI:Migraine domain scores are expressed as percentages.

**Table 2 neurolint-17-00202-t002:** Test–retest reliability of the Arabic WPAI:Migraine domain scores using a two-way mixed-effects model, absolute agreement, single measurement (n = 47).

Domain	Test Mean (SD)	Retest Mean (SD)	Mean Difference (95% CI)	ICC (95% CI, *p* Value)
Absenteeism	15.54 (22.83)	11.70 (16.75)	4.03 (−3.80–11.87)	0.76 (0.45–0.90, <0.001 *)
Presenteeism	48.09 (29.17)	41.48 (29.31)	5.77 (−3.24–14.78)	0.84 (0.65–0.93, <0.001 *)
Overall work productivity loss	39.42 (25.20)	36.71 (25.00)	2.66 (−7.89–13.21)	0.68 (0.27–0.86, 0.004 *)
Activity impairment	58.29 (28.06)	49.77 (26.86)	7.91 (1.21–14.61)	0.812 (0.65–0.90, <0.001 *)

SD = standard deviation; CI = confidence interval; ICC = intraclass correlation coefficient using two-way mixed-effect model; * Significant at *p* < 0.05.

**Table 3 neurolint-17-00202-t003:** Concurrent validity: correlation between WPAI:Migraine domain scores and MIDAS total score (n = 47).

WPAI:Migraine Domain	Pearson’s r	95% CI (*p* Value)	Spearman’s ρ	95% CI (*p* Value)
Absenteeism	0.356	0.07–0.59, 0.018 *	0.522	0.26–0.71, <0.001 *
Presenteeism	0.408	0.13–0.62, 0.005 *	0.526	0.27–0.71, <0.001 *
Overall work impairment	0.149	−0.16–0.48, 0.336	0.294	−0.01–0.55, 0.053
Activity impairment	0.452	0.19–0.65, 0.001 *	0.576	0.34–0.74, <0.001 *

WPAI = Work Productivity and Activity Impairment; MIDAS = Migraine Disability Assessment. * Significant at *p* < 0.05.

**Table 4 neurolint-17-00202-t004:** Known-group validity: WPAI:Migraine domain scores by MIDAS disability grade (n = 82) using one analysis of variance with Least Significant Difference post hoc analysis.

WPAI:Migraine Domain	Grade I	Grade II	Grade III	Grade IV	F	df	*p*-Value
Absenteeism	0 (0)	1.79 (3.57)	1.85 (3.75)	21.72 (25.15)	1.55	3	0.040 *
Presenteeism	0 (0)	55.00 (31.09)	36.67 (26.93)	57.59 (24.74)	0.30	3	<0.001 *
Overall work impairment	0 (0)	54.30 (31.57)	35.70 (28.77)	43.09 (21.37)	3.74	3	0.019 *
Activity impairment	2.50 (5.00)	52.50 (26.30)	27.78 (22.79)	58.67 (20.97)	11.62	3	<0.001 *

Values are mean (SD). WPAI = Work Productivity and Activity Impairment; MIDAS = Migraine Disability Assessment; SD = standard deviation. Higher scores indicate greater impairment. * Significant at *p* < 0.05; df: degree of freedom.

## Data Availability

All data generated or analyzed during this study are included in this published article and its [Appendix A]. Further inquiries can be directed to the corresponding author.

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
