# Peer review of "Psychometric Validation of the Arabic Version of the WPAI:Migraine Questionnaire in Patients with Migraine"

_2035-8377, 2025, doi:10.3390/neurolint17120202_

Round 1
Reviewer 1 Report
Comments and Suggestions for Authors
This a validation study of the WPAI for people with migraine in the arabic language. Although WPAI is already validated in Arabic, the authors have re validated it , with the focus in migraine , and by slightly adapting the already existing Arabic WPAI. The methodology used and the statistical analysis are adequate. The manuscript is well written . I only have some minor comments :
- Presenteism and absenteism : their explanations are provided later on the manuscript and not where they first appear.
- How did the authors decide the study sample? Was a power analysis performed?
- Please describe how the participants completed MIDAS and WPAI via zoom. In case someone read these to them and they chose an answer, this may create a bias since when somebody completes a PRO by himself may choose differently from when someone is in front of him and checks every answer the participant gives. In other words, the in-person setting can lead to discrepancies in the data because the patient's report can be influenced by the immediate interaction with the provider. This may be a limitation worth discussed. The following might be helpfull : https://doi.org/10.2147/PROM.S178344
Overall, the methodology of the manuscript is sound . The authors clearly present their findings . Hopefully, this tool may enable further studies into the productivity loss of the Arabic population.
Reviewer 2 Report
Comments and Suggestions for Authors
The authors conducted a cross-sectional psychometric validation study to assess the Arabic version of the WPAI:Migraine questionnaire, comparing it with the validated Arabic version of the Migraine Disability Assessment Scale. The study included 82 patients with episodic or chronic migraine. The test-retest reliability, concurrent validity, and known-groups validity were evaluated. Given the high prevalence of migraine and its impact on work productivity and overall activity, the study addresses a clinically relevant topic.
The manuscript is well organized. The Introduction section is well-structured and justifies the aim of the study. The material and methods are described in sufficient details to ensure transparency and reproducibility of the results. I have the following comments regarding the presentation of the results:
Lines 180–187 repeat in detail the information already presented in Table 1. I suggest that the authors shorten this section to avoid redundancy. In addition, in the second row of Table 1, the label should read “Female sex, n (%)”.
For greater transparency and reproducibility, the authors should provide additional statistical information in Table 2, including the ICC confidence interval, the ICC model, and the sample size (n). The table caption should also specify the type of statistical test used, for example: “two-way mixed-effects model with …”
For clarity, the authors should present the correlation coefficients and their corresponding p-values in Table 3 in separate columns. For the Pearson correlation coefficients, the confidence intervals should also be included in the table. The sample size should also be written. Additionally, the text (lines 217 – 220) should discuss not only statistical significance but also the strength and direction of the correlations.
In Table 4, the authors should provide additional statistical information (F, df) and indicate which group(s) differ statistically from others, rather than reporting only the level of significance. The table caption should also specify the type of statistical analysis and the post hoc test used.
Reviewer 3 Report
Comments and Suggestions for Authors
In the Introduction, the statement about presenteeism being more important than absenteeism is crucial. Consider citing a systematic review or a key study that establishes this in the migraine literature to strengthen this claim.
To make introduction more impactful, you could briefly state the expected utility of the tool, e.g., "...to assess the impact of migraine on work productivity... thereby supporting both clinical management and health economic evaluations."
The adaptation from WPAI:GH (General Health) by direct substitution may overlook subtle semantic or cultural nuances specific to migraine; broader regional piloting is advised for future studies.
Though adequate for initial psychometric analysis, the clinic-based sample (n = 82) primarily included employed, highly educated patients, limiting diversity. Selection from a tertiary headache clinic may bias toward more severe or chronic migraine cases, restricting broader population generalizability. Data was collected at one tertiary hospital, which may further limit external validity and perspectives from other settings (e.g., primary care, non-urban environments).
Regarding retest procedure, only 43 of 82 participants completed the two-week retest, which is a substantial drop-off and could introduce retest bias. The reasons for attrition and its impact on results warrant further investigation.
The ICC range is given as 0.68–0.84. It would be helpful to specify which domains correspond to the lower and higher ends of this range for clarity (e.g., "ranging from 0.68 for overall work productivity loss to 0.84 for presenteeism").
The sample is notably skewed toward high disability, with 64% of patients in MIDAS Grade IV (severe). While this reflects a tertiary care population, it is important to discuss how this might affect the instrument's performance. For instance, it may be exceptionally sensitive in detecting impairment in severely affected groups, but its performance in populations with milder migraine (e.g., in primary care or community settings) could be an area for future research.
The study design does not allow for assessment of responsiveness to change, an important feature when using PROMs to evaluate interventions or longitudinal trends. Future longitudinal studies are needed to confirm the instrument’s utility for repeated measures.
The adaptation relied on substituting “general health” for “migraine” without full forward-backward translation or formal linguistic validation. Nuances specific to migraine language use may not be fully captured in different Arabic-speaking regions.
According to Table 3, the non-significant result for "Overall work impairment" with Pearson's correlation is interesting. It might be worth a single sentence in the results or discussion speculating why this might be, given that its components (absenteeism and presenteeism) were significantly correlated. This could be due to the composite nature of the score or the specific sample distribution.
In Figure 1, ensure the axes are clearly labeled in the final version.
Comments on the Quality of English Language
The English could be improved to more clearly express the research.
Round 2
Reviewer 3 Report
Comments and Suggestions for Authors
The authors have well addressed my comments.
Comments on the Quality of English Language
The English could be improved to more clearly express the research.